# OpenSRH: optimizing brain tumor surgery using intraoperative stimulated Raman histology

**Cheng Jiang**[1*]    **Asadur Chowdury**[1*]    **Xinhai Hou**[1*]
**Akhil Kondepudi**[1]    **Christian W. Freudiger**[2]    **Kyle Conway**[1]
**Sandra Camelo-Piragua**[1]    **Daniel A. Orringer**[3]    **Honglak Lee**[1]    **Todd C. Hollon**[1]

[1]University of Michigan    [2]Invenio Imaging    [3]New York University    [*]Equal Contribution

{chengjia, achowdur, xinhaih, tocho}@umich.edu    https://opensrh.mlins.org

## Abstract

Accurate intraoperative diagnosis is essential for providing safe and effective care during brain tumor surgery. Our standard-of-care diagnostic methods are time, resource, and labor intensive, which restricts access to optimal surgical treatments. To address these limitations, we propose an alternative workflow that combines stimulated Raman histology (SRH), a rapid optical imaging method, with deep learning-based automated interpretation of SRH images for intraoperative brain tumor diagnosis and real-time surgical decision support. Here, we present *OpenSRH*, the first public dataset of clinical SRH images from 300+ brain tumors patients and 1300+ unique whole slide optical images. OpenSRH contains data from the most common brain tumors diagnoses, full pathologic annotations, whole slide tumor segmentations, raw and processed optical imaging data for end-to-end model development and validation. We provide a framework for patch-based whole slide SRH classification and inference using weak (i.e. patient-level) diagnostic labels. Finally, we benchmark two computer vision tasks: multiclass histologic brain tumor classification and patch-based contrastive representation learning. We hope OpenSRH will facilitate the clinical translation of rapid optical imaging and real-time ML-based surgical decision support in order to improve the access, safety, and efficacy of cancer surgery in the era of precision medicine. Dataset access, code, and benchmarks are available at https://opensrh.mlins.org.

## 1  Introduction

The optimal surgical management of brain tumors varies widely depending on the underlying pathologic diagnosis [1]. Surgical goals range from needle biopsies (e.g. primary central nervous system lymphoma [2]) to supramaximal resections (e.g. diffuse gliomas [3]). A major obstacle to the precision care of brain tumor patients is that the pathologic diagnosis is usually *unknown* at the time of surgery. For other tumor types, such as breast or lung cancer, diagnostic biopsies are obtained prior to definitive surgical management, which provides essential clinical information used to inform the goals of surgery. Routine diagnostic biopsies in neuro-oncology are not feasible due to high surgical morbidity and the potential for permanent neurologic injury. Consequently, the importance of *intra*operative pathologic diagnosis in brain tumor surgery has been recognized for nearly a century[4].

Unfortunately, our current intraoperative pathologic techniques are time, resource, and labor intensive [7, 8]. Conventional diagnostic methods, including frozen sectioning and cytologic preparations, require an extensive pathology infrastructure for tissue processing and specimen analysis by a board-

36th Conference on Neural Information Processing Systems (NeurIPS 2022) Track on Datasets and Benchmarks.

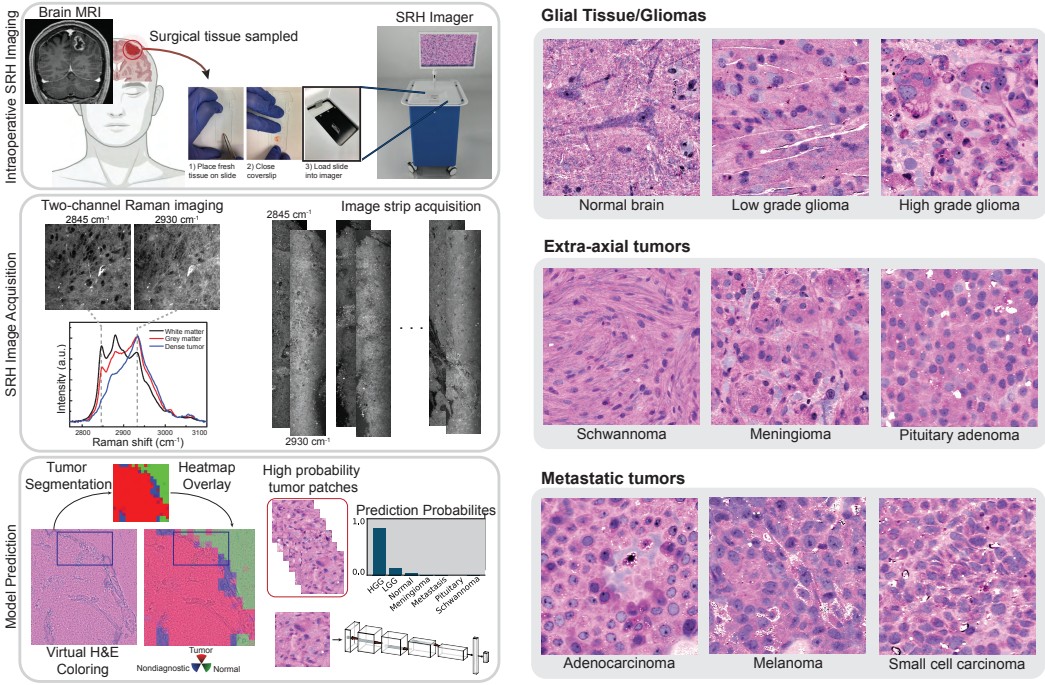

Figure 1: **Left**, A patient with a newly diagnosed brain lesion undergoes a surgery for tissue diagnosis and/or tumor resection. The tumor specimen is sampled from the patient's tumor and directly loaded into a premade, disposable microscope slide. The specimen is placed into the SRH imager for rapid optical imaging. SRH images are acquired sequentially as strips at two Raman shifts, 2845 cm$^{-1}$, and 2930 cm$^{-1}$. The size and number of strips to be acquired are set by the operator who defines the desired image size. Standard image sizes range from 1-5mm$^2$ and image acquisition time ranges from 30 seconds to 3 minutes. The strips are edge clipped, field flattened and co-registered to generate whole slide SRH images. Images can be colored using a custom virtual H&E colorscheme for pathologic review [5]. The whole slide image is divided into non-overlapping $300\times300$ pixel patches and each patch undergoes a feedforward pass through a previously trained tumor segmentation model [6] to segment the patches into tumor regions, normal brain, and nondiagnostic regions. The tumor patches are then used for both training and inference to predict the patient's brain tumor diagnosis. **Right**, Examples of virtually-colored SRH images from brain tumor diagnoses included in OpenSRH. We include a diversity of tumor diagnoses that cover the most common brain tumor subtypes.

certified neuropathologist [9]. While the conventional pathology workflow with board certified neuropathologist interpretation has a diagnostic accuracy between 86 - 96% [5], the pathologist workforce in the US declined in absolute and population-adjusted numbers by nearly 20% between 2007-2017 [10]. This decline has unevenly affected neuropathology, with a 40% fellowship vacancy rate and it is projected to worsen [11]. The number of medical centers performing brain tumor surgery outnumbers board-certified neuropathologists, reducing patient access to expert intraoperative consultation and, consequently, optimal surgical management.

An ideal system for surgical specimen analysis and intraoperative tumor diagnosis would be accessible, fast, standardized, and accurate. An intraoperative pathology system requires, at minimum, (1) a data/image acquisition modality and (2) a diagnostic interpretation. Conventional intraoperative pathology uses light microscopy interpreted by a neuro-pathologist (>20-30 minutes). We propose an innovative diagnosis system that uses a rapid (2-3 minutes, >10× speedup), label-free optical histology method, called stimulated Raman histology (SRH), combined with deep learning-based interpretation of fresh, unprocessed surgical specimens. We have previously demonstrated the feasibility of large-scale clinical SRH imaging [5] and the use of deep neural networks for SRH image classification of brain tumor patients [6, 12, 13]. These studies demonstrate the potential

for AI-based diagnosis and interpretation of SRH images to better inform brain tumor surgery and provide personalized surgical goals in the era of precision medicine.

Here, we seek to facilitate this area of active research by releasing *OpenSRH*, a collection of intraoperative SRH data, including raw SRH acquisition data, processed high-resolution image patches for model development, virtually-stained whole slide images, semantic segmentation of tumor regions, and full intraoperative diagnostic annotations. OpenSRH is the first and only publicly available dataset of any human cancer imaged using optical histology. We release the OpenSRH dataset with the intention to foster translational AI research within the field of precision oncology. The main contributions of this work are:

1. **OpenSRH dataset**: We curate and open-source the largest dataset of intraoperative SRH images with pathologic annotations to facilitate the development of innovative machine learning solutions to improve brain tumor surgery.

2. **Classification benchmarks**: We benchmark performance for patch-based histologic brain tumor classification across multiple tumor types, computer vision architectures, and transfer learning methods.

3. **Contrastive representation learning benchmarks**: We evaluate both self-supervised and weakly supervised patch contrastive learning methods for SRH representation learning. Contrastive learning methods are evaluated using linear evaluation protocols and benchmarked as a model pretraining strategy.

## 2   Background

**Stimulated Raman Histology**   SRH is based on Raman scattering. Raman scattering occurs when incident photons on a media either gain or lose energy when scattered (i.e. inelastic scattering), shifting the frequency/wavenumber of the scattered photons. This Raman shift can be measured to characterize the biochemical composition of both inorganic and organic materials using narrow-band laser excitation and a spectrometer [14]. A major limitation of using spontaneous Raman scattering for biochemical analysis is that the Raman effect is weak compared to elastic scattering. Therefore, long acquisition times (> 30 minutes) and spectral averaging are required to obtain representative biochemical spectra. Stimulated Raman scattering (SRS) microscopy was discovered in 2008 as a highly sensitive, label-free biomedical imaging method [15]. Rather than acquiring broad-band spectra, SRS microscopy uses a second laser excitation source to achieve non-linear amplification of narrow-band Raman spectral regions that correspond to specific molecular vibrational modes (see Figure 1). SRS images can then be generated at specific narrow-band Raman wavenumbers. Translational research led to the development of a fiber-laser-based SRS imaging system that could be used at the patient's bedside to generate rapid histologic images of fresh surgical specimens, called SRH [5, 16, 17]. A major advantage of SRH over other histologic imaging methods is that image contrast is generated by the intrinsic biochemical properties of the tissue only and does not require any tissue processing, staining, dyeing, or labelling (i.e. label-free).

**ML applications for SRH**   Unlike conventional intraoperative histology with light microscopy, SRH provides high-resolution *digital* images that can be used directly for downstream ML tasks. Whole slide image digitization of frozen or paraffin-embedded tissue is slow and memory intensive, presenting a major bottleneck for its routine use in intraoperative histology, and clinical medicine in general [18]. Previous studies showed that SRH plus shallow ML models can be used to detect and quantify tumor infiltration in adult and pediatric fresh surgical specimens [5, 19, 20, 21]. We subsequently demonstrated that SRH combined with convolutional neural network architectures can be used for intraoperative diagnostic decision support [6, 12, 13]. These preliminary studies, while demonstrating the feasibility of applying deep architectures to SRH, did not include rigorous hyperparameter tuning, explicit representation learning, or ablation studies to optimize model performance. Moreover, all previous studies required manual annotations, including dense patch-level annotations [6], for model training.

# 3   Related Work

To date, no SRH datasets are publicly available. The work most directly related to OpenSRH comes from digital and computational pathology research. The Cancer Genome Atlas (TCGA) and The Cancer Imaging Atlas (TCIA) include a large repository of digitized histopathology slides processed using hematoxylin and eosin (H&E) staining. Many studies have used this dataset for image classification tasks across several cancer types, including, but not limited to, lung [22], gastrointestinal [23], prostate [24, 25], brain [26], and pan-cancer studies [23, 27, 28]. Another related histopathology dataset comes from the CAMELYON16 research challenge [29]. The challenge is to detect lymph node metastases in women with breast cancer. Digital pathology remains an active area of research in precision oncology. However, ML applications in digital pathology are mainly applied to postoperative tissue assessment and do not play a major role in informing cancer surgery.

One application of SRH is the detection of tumor infiltration in real-time to improve the extent of tumor resection and reduce residual tumor burden. Real-time SRH-based tumor delineation has been studied in sinonasal/skull base cancers [13, 30, 31] and diffuse gliomas [20, 32]. OpenSRH provides the necessary dataset to explore this topic for multiple brain tumor types, including metastatic tumors and extra-axial tumors, such as meningiomas (Figure 1).

**Overall Need**   High-quality, public, biomedical datasets with expert annotations are rare. Moreover, the clinical significance of some existing datasets is unclear due to the lack of a roadmap for clinical translation [33]. We believe that OpenSRH has the potential to address a currently unmet clinical need of improving cancer surgery in order to advance precision oncology, both in the US and globally [7]. OpenSRH can address a pressing and significant clinical problem, while having high translational potential because, as previously mentioned, an ideal system for intraoperative tumor specimen evaluation should be:

1. **Accessible:** SRH imaging systems are FDA-approved and commercially available for intraoperative imaging

2. **Fast:** imaging acquisition time and time-to-diagnosis is $10\times$ faster than the current standard-of-care H&E histology

3. **Standardized:** SRH image acquisition is invariant to patient demographic features, clinical workforce, and geographic location

4. **Accurate:** preliminary results [6, 13] and diagnostic performance benchmarks (see Figure 4) are on par with the pathologist-based interpretation of H&E histology

# 4   Data Description

**Patient population**   Patients were consecutively and prospectively enrolled for intraoperative SRH imaging. This study was approved by Institutional Review Board (HUM00083059). Informed consent was obtained for each patient prior to SRH imaging and approved the use of tumor specimens for research and development. All patient health information (PHI) are removed from all OpenSRH data. The inclusion criteria are (1) patients with planned brain tumor or epilepsy surgery at Michigan Medicine (UM), (2) subjects or durable power of attorney able to give informed consent, and (3) subjects in whom there was additional specimen beyond what was needed for routine clinical diagnosis.

**SRH imaging**   Intraoperative SRH imaging and data processing workflow can be found in Figure 1. A small tumor specimen ($3\times3$ mm$^3$) is placed into a premade microscope slide, which is then loaded into the commercially available NIO Imaging System (Invenio Imaging, Inc.) for SRH imaging. The tissue is then excited with a dual-wavelength fiber laser source, which provides spectral access to Raman shifts in the range of 2800 cm$^{-1}$ to 3130 cm$^{-1}$. SRH images are acquired at two Raman shifts: 2845 cm$^{-1}$ highlights lipid-rich regions and 2930 cm$^{-1}$ highlights DNA and protein-rich regions [6]. The images are acquired sequentially as 0.5 mm wide strips, stitched together and the two image channels are co-registered to generate the final whole slide image. The co-registration between the two image channels is performed using discrete Fourier transform. A virtual H&E colorscheme [5] can be applied to SRH images for clinician review, but is not used for model development.

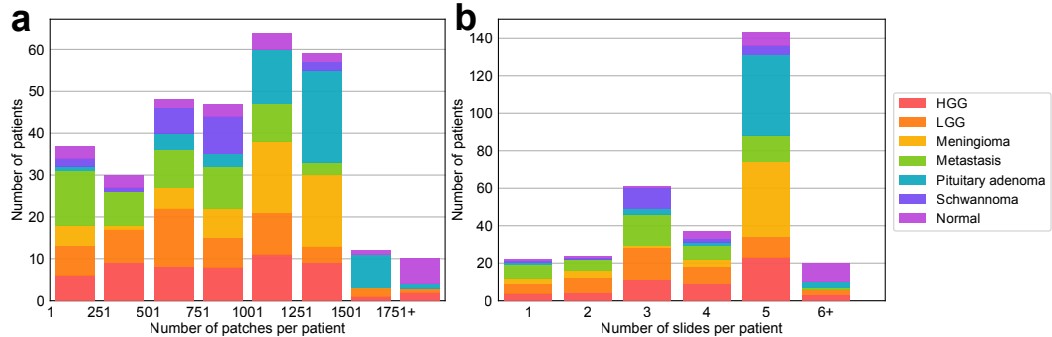

Figure 2: Histogram for the number of patches and slides per patient. **a** shows the total number of patches varies across the patients and, **b** shows most patients have 5 slides. The difference between these two distributions may be caused by specimen size, non-diagnostic regions, surgeon preference, etc. HGG, high grade glioma; LGG, low grade glioma.

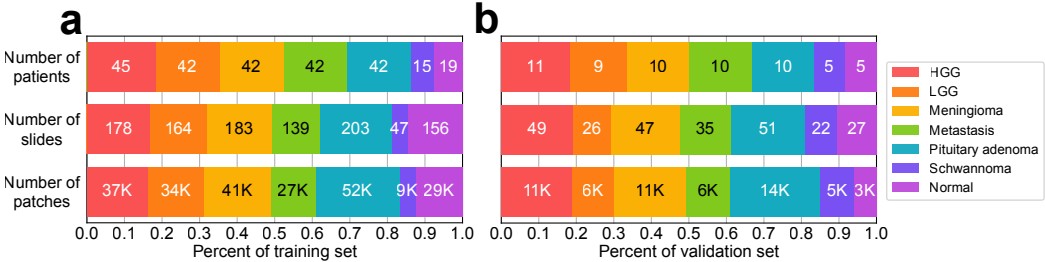

Figure 3: Bar chart for the number of patients, slides and patches for each diagnostic class. Validation set was randomly selected and contains approximately 20% of patients in OpenSRH (60/307 patients). Training and validation sets have approximately equivalent class distributions.

**Image preprocessing** Image processing starts by applying a sliding window over the two-channel image to generate $300 \times 300$ pixel non-overlapping patches. A third channel is obtained by performing a pixel-wise subtraction from the two registered channels (2930cm$^{-1}$ - 2845cm$^{-1}$), which highlights the nuclear contrast and cellular density of the tissue [19]. The third channel is concatenated depth-wise to generate a final three-channel patch for model training and inference. Each patch then undergoes a feedforward pass through a pretrained segmentation model to classify the patch into tumor, normal brain, or non-diagnostic tissue [12]. The model was trained using manually labelled patches. The segmentation prediction for each patch is released as part of the OpenSRH dataset.

**Dataset breakdown** OpenSRH consists of 307 patients. A total of 304 patients underwent intra-operative SRH imaging and three patients had postmortem specimen collection. We strategically selected the most common brain tumor types to be included in OpenSRH. The included brain tumor diagnoses cover more than 90% of all newly diagnosed brain tumors in the US [34]. OpenSRH includes a diversity of brain tumor types, including primary brain tumors (high-grade gliomas, low-grade gliomas), secondary brain tumors (metastases), and extra-axial tumors (meningiomas, schwannomas, pituitary adenomas). A panel of patch samples is included in Figure 1. The dataset is randomly divided into training (247 patients) and validation set (60 patients, about 20%). Figure 3 shows a distribution of the number of patients, slides, and patches per class in the training and validation set. Technical details of the data release and companion source code are in Appendix A.

## 5 Histologic brain tumor classification benchmarks

In this section, we present the results of the baseline multiclass brain tumor classification task. We aim to benchmark the results for common training strategies. We investigate the value of

transfer learning/pretraining from natural image datasets, specifically ImageNet [35], for improving classification performance. The value of transfer learning for SRH [6] and medical imaging, in general [36], remains an active area of research. We selected representative models from the two most competitive computer vision architectures: convolutional neural networks (ResNet50 [37]) and vision transformers (ViT-S[38, 39]). These architectures were selected because they contain a similar number of parameters ($\sim$23.5 million for ResNet50, $\sim$21.7 million for ViT-S).

## 5.1 Training protocol

**ResNet50**   In the ResNet50 architecture, we changed the output dimension of the last layer to 7 for our experiments. We used a batch size 96 and trained on 300×300 images with horizontal and vertical flipping of probability 0.5 as augmentations. We used categorical cross-entropy loss and AdamW optimizer [40] with $\beta_1 = 0.9$, $\beta_2 = 0.999$, and a weight decay of 0.001. The initial learning rate was 0.001, with a step scheduler with a decay rate $\gamma = 0.5$ every epoch. We trained for 20 epochs with two Nvidia RTX 2080Ti GPUs. Training wall time was $\sim$9.5 hours for each experiment.

**ViT-S**   ViT training protocol was adjusted based on previously published results[38, 39]. In addition to the augmentations in the ResNet50 protocol, we also resized the image to 224×224 to fit the standard ViT-S model and ImageNet pretrained weights [41]. We used AdamW as the optimizer, with the same parameters in ResNet50. The initial learning rate was 1E-4, with a cosine learning rate scheduler. First 20% of training steps were set as the linear warm-up stage to increase training stability. We trained 20 epochs with a batch size of 256 for 9 hours using the same GPU resources mentioned above. Detailed training parameters are included in Appendix C.

## 5.2 Prediction aggregation and benchmark metrics

Patch-level predictions from the same whole slide or patient need to be aggregated to generate a slide- or patient-level prediction.   We aggregated patch-level logits after softmax using average pooling to compute slide and patient-level prediction. We preferred using average pooling over hard patch voting to retain the full patch-level model predictions during slide- or patient-level inference. Model performance evaluation metrics include top-1 accuracy, mean class accuracy (MCA), and mean average precision (MAP). Additional classification metrics including top-2 accuracy and false negative rate (tumor vs. normal) can be found in Appendix D.

## 5.3 Experimental results

Patch- and patient-level results can be found in Table 1. Patient-level metrics are generally higher than patch-level metrics. Patch-level prediction errors can be mitigated through the average pooling aggregation function. In our preliminary benchmark, ResNet50 achieved overall better performance than ViT-S  (e.g., by 7.2 patch accuracy and 5.6 patient accuracy). A potential reason is due to ViT requiring large-scale image datasets on the scale of ImageNet21K or JFT300M [38] to overcome low inductive bias.  Insufficient pretraining is known to result in worse performance compared to convolutional neural networks (CNNs).  We did observe improved  patch-level performance when using ImageNet pretraining  (2.1 for ResNet50 and 6.5 for ViT-S at patch accuracy).  In general, pretraining was more beneficial to ViT than ResNet50. We believe vision transformers may outperform CNNs with data efficient pretraining. Figure 4 summarizes the patient-level confusion matrix. Both models had similar diagnostic errors differentiating HGG and LGG, a known challenging diagnostic task for pathologists and computer vision models [5]. Metastatic tumors have diverse histologic features (see Figure 1) that can result in diagnostic errors across multiple classes [6]. From the confusion matrices in figure 4, it is important to note that we can also observe some false negatives in the model prediction (tumor vs. normal). Additional metrics on false negative rate for these experiments are also included in appendix D.

## 6   Contrastive representation learning benchmark

Our previous studies demonstrate that contrastive representation learning is well suited for patch-based representation learning [42]. The focus for this section is to investigate the effectiveness of contrastive learning strategies for OpenSRH. We used both unsupervised contrastive learning (SimCLR [43]) and

| Backbone | Pretrain | Patch | | | Patient | | |
|---|---|---|---|---|---|---|---|
| | | Accuracy | MCA | MAP | Accuracy | MCA | MAP |
| ResNet50 | Random | 84.4 (0.4) | 83.8 (0.5) | 89.5 (0.5) | **90.0 (0.0)** | **91.4 (0.0)** | 92.8 (0.2) |
| ResNet50 | ImageNet | **86.5 (0.4)** | **85.6 (0.3)** | **91.2 (0.3)** | 88.9 (0.8) | 90.5 (0.6) | **94.0 (0.1)** |
| ViT-S | Random | 77.2 (0.5) | 76.8 (0.8) | 82.3 (0.5) | 85.0 (1.4) | 87.2 (1.1) | 93.2 (0.4) |
| ViT-S | ImageNet | 83.7 (0.5) | 82.7 (0.9) | 88.8 (0.1) | 88.9 (0.8) | 90.5 (0.6) | **93.9 (0.4)** |

Table 1: Classification benchmarks for ResNet50 and ViT-S. Pretrain refers to the pretraining strategy. Each experiment included three random initial seeds. Mean value and standard deviation (in parentheses) for each metric are reported here. The full table including false negative rates and slide-level metrics can be found in the Appendix D.

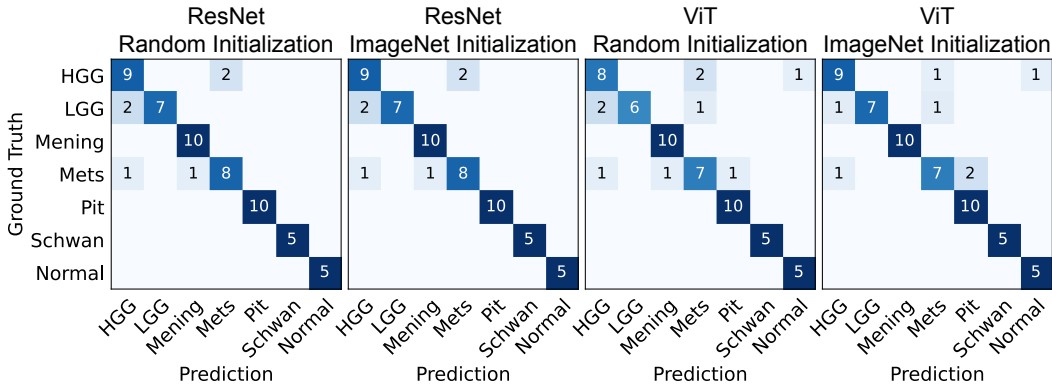

Figure 4: Patient-level confusion matrices for the four different training strategies on the validation set. Most of the errors occurred in the HGG, LGG, and metastasis classes. Only seed 1 is shown here, other seeds are included in Appendix D. Mening, meningioma; Mets, metastasis; Pit, pituitary adenoma; Schwan, schwannoma.

supervised contrastive learning (SupCon [44]) on ResNet50 and ViT-S architectures. The contrastive loss for SimCLR aims to solve the pretext task of instance discrimination. The model is trained to identify two augmented positive pairs of the same image from other images in a minibatch. SupCon loss has the similar training objective but aims at optimizing class discrimination. All images from the same class are treated as positive instances and other images are negative instances. Our general contrastive learning workflow uses SimCLR and SupCon as a representation learning strategy on our dataset followed with a linear evaluation protocol. We compared contrastive representation learning methods with ImageNet pretraining.

## 6.1 Training and evaluation protocol

**Training protocol** For both SimCLR and SupCon methods, we use the same protocol except for the loss function. We applied ResNet50 and ViT-S with a linear projection head to project the image representation to a low dimensional hypersphere (128 for ResNet50, 24 for ViT-S) to compute the contrastive loss. The data augmentation strategy followed [43]: a composition of multiple augmentations including flipping, color jittering, and Gaussian blur (for details, see Appendix C). We use AdamW optimizer for both models and same parameters as in Section 5.1. Different learning rates (1E-3 for ResNet50 and 5E-4 for ViT-S) were adopted for each model. We trained using a batch size 224 for ResNet50 and 512 for ViT-S for 40 epochs. We use linear warmup for the first 10% epochs and cosine decay scheduler for ViT only. Detailed protocols are included in Appendix C.

**Evaluation protocol** To evaluate the learned image representations, we followed a standard linear evaluation protocol [43, 45, 46, 47], where a linear classifier is trained on top of the frozen pretrained backbone. We consider the same evaluation metrics and aggregation function as in Section 5.2. Apart from the classification metrics, we performed qualitative evaluation of our learned representations

through t-distributed stochastic neighbor embedding (tSNE) [48] in Figure 5. Additional fine-tuning protocols and results are included in Appendix F.

## 6.2 Experimental results

Results of linear evaluations could be found in Table 2. Self-supervised representation learning with SimCLR was able to achieve a patient-level accuracy of 85.6 for ResNet50 and 78.3 for ViT-S. These results demonstrate improvement over our previous self-supervised classification performance [49]. Self-supervised contrastive representation learning using OpenSRH outperforms ImageNet transfer learning for patch-based metrics (e.g., by 3.2 for ResNet50 and 2.3 for ViT-S). These results emphasize the large domain gap between natural images and SRH optical images [50, 36]. Similar to other computer vision tasks, optimal representation learning can be achieved with additional supervision. SupCon outperforms both ImageNet pretraining and SimCLR in patch-based metrics. Linear evaluation showed an overall increase of 4.4 and 8.4 in patient-level accuracy between SupCon and SimCLR for ResNet50 and Vit-S, respectively. Interestingly, the patient-level metrics for pretrained ViT-S were prominently high, while the patch-level metrics were comparatively worse. We believe these results may be due to a simple soft voting aggregation of patch-level predictions. This opens the question for better (learnable) aggregation functions for SRH images. The tSNE plot in Figure 5 was consistent with our patch-based evaluation metrics for both models, where SupCon showed more discrete image representations.

| Backbone | Methods | Patch | | | Patient | | |
|---|---|---|---|---|---|---|---|
| | | Accuracy | MCA | MAP | Accuracy | MCA | MAP |
| ResNet50 | ImageNet | 68.3 (0.0) | 67.9 (0.0) | 72.9 (0.1) | 80.0 (0.0) | 82.9 (0.0) | 88.8 (0.1) |
| ResNet50 | SimCLR | 79.1 (0.4) | 78.9 (0.4) | 84.2 (0.6) | 83.9 (1.0) | 86.1 (0.9) | 92.4 (0.1) |
| ResNet50 | SupCon | **87.5 (0.3)** | **86.8 (0.3)** | **91.5 (0.5)** | **90.0 (0.0)** | **91.4 (0.1)** | **94.6 (0.5)** |
| ViT-S | ImageNet | 71.8 (0.1) | 71.1 (0.1) | 77.1 (0.1) | 88.3 (0.0) | 89.8 (0.0) | 93.9 (0.0) |
| ViT-S | SimCLR | 76.8 (0.5) | 76.3 (0.5) | 82.5 (0.3) | 80.0 (1.7) | 83.0 (1.3) | 92.3 (0.0) |
| ViT-S | SupCon | 81.4 (0.2) | 80.2 (0.3) | 85.6 (0.5) | 87.8 (1.0) | 89.4 (0.7) | **94.0 (0.4)** |

Table 2: Linear evaluation protocol results for contrastive representation learning. Each experiment included three random initial seeds. Mean value and standard deviation (in parentheses) for each metric are reported here. The full table including false negative rates and slide-level metrics can be found in the Appendix E.

## 7 Limitations, Open Questions, and Ethical Consequences

OpenSRH contains data collected from a single institution. While SRH imagers have standardized settings, different operating room workflows, tumor sampling strategies, and surgeons may produce SRH data distribution shifts. Moreover, while our OpenSRH does contain the most common brain tumor types, rare tumor classes are not included. This is a limitation because rare tumor diagnosis is one of the contexts in which ML-based diagnostic decision support can be most beneficial to clinicians. We intend to include multicenter data with additional tumor rare classes in future releases of OpenSRH.

There are many open questions for the machine learning community that can be explored through OpenSRH. The most important questions are given below:

- **Domain adaptation.** Many domain adaptation literature uses datasets that have very small domain gaps such as MNIST [51], SVHN[52], Office 31[53], or datasets that are artificially crafted or generated, such as Adaptiope [54] or DomainNet [55]. OpenSRH can be combined with existing H&E dataset such as TCGA, to create a large-scale benchmark for domain adaptation, with intrinsic pathologic features captured using different imaging modalities.
- **Multiple instance learning.** Besides our patch-based classification workflow, multiple instance learning may be a good strategy for histopathology analysis [56]. By removing patch labels in OpenSRH, histologic classification becomes a generic multiple instance learning task. A model can learn to select the important patches with only slide-level labels. Our current patch-level annotation can be used as a ground truth to interpret instance-level predictions from multiple instance learning paradigms.

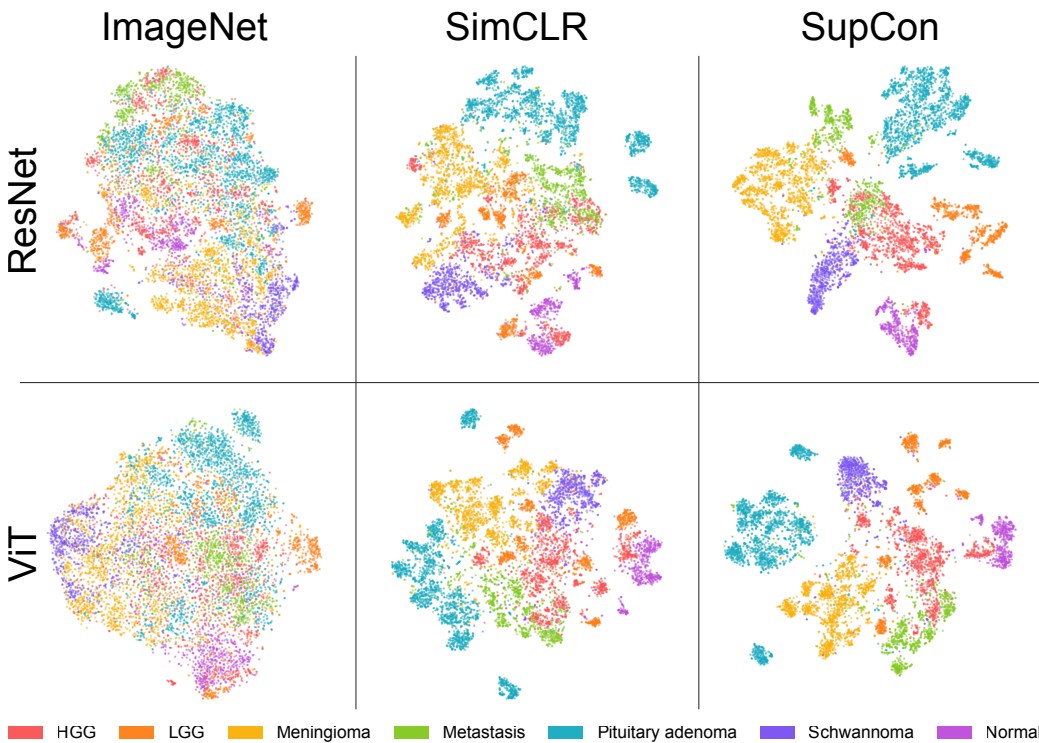

Figure 5: Patch-level SRH representations of validation images. Latent feature vectors are colored by tumor class labels. ImageNet pretraining fails to represent discriminative SRH image features. SimCLR shows discernable SRH feature learning capabilities, while improved class separation can be learned with SupCon. Tumor classes with similar SRH histologic features tend to show similar feature representations, such as HGG/LGG and HGG/Metastasis. A single seed is shown here. Figure best viewed in color.

- **Aggregation of patch-level predictions.** We have relied on individual patch-level predictions and average pooling as a general method for whole slide inference. However, this strategy is limited because it does not account for discriminative heterogeneity within whole slide images. Expectation maximization [57], clustering [58, 28], attention [56, 59], and other multiple instance learning strategies [60] have been proposed as learnable aggregation functions, but questions related to scalability, training efficiency, and data domain differences remain open.

- **Self-supervised learning.** Self-supervised learning and contrastive learning methods have been explored using histology images, but the effectiveness of different augmentation strategies has not been studied. Our preliminary experiments indicate that augmentations used for natural image self-supervised representation learning are sub-optimal. It remains an open question whether domain specific augmentation would improve self-supervised learning performance.

- **Data efficient training.** It is known that ViTs require large image datasets on the scale of ImageNet21K or JFT300M, and insufficient pretraining can result in inferior performance [38]. Acquiring these large supervised datasets is currently infeasible in medical imaging. In addition to low inductive bias, ViTs demonstrate better interpretability compared to CNNs, and their clinical adoption can improve reliability and physician's trust in AI-assisted diagnosis. By demonstrating a performance gap between CNNs and ViTs, our OpenSRH benchmarks encourage research in data efficient training of ViTs suited for medical imaging.

Lastly, we have ensured that all patients consented to release a portion of their tumor for research. There is minimal additional risk for patients because samples are collected from tumors removed as part of the standard patient care, and their personal health information is protected in OpenSRH.

OpenSRH is released to promote translational AI research. The dataset, algorithms and benchmarks discussed in the paper are for research purposes only.

## 8 Conclusion

In this work, we introduce OpenSRH, an intraoperative brain tumor dataset of SRH, a rapid, label-free, optical imaging method. OpenSRH contains both raw SRH acquisition data and processed high-resolution image patches for model development using diagnostic annotations from expert neuropathologists. OpenSRH is the first and only publicly available dataset of human cancers imaged using optical histology. We benchmark classification performance for histologic brain diagnosis across the most common brain tumor types. We also provide benchmarks for self-supervised and weakly supervised contrastive representation learning. We release the OpenSRH dataset with the intention to promote translational AI research within the field of precision oncology and optimize the surgical management of human cancers.

### Acknowledgements, Disclosure of Funding and Competing Interests

We would like to thank Karen Eddy, Lin Wang, Andrea Marshall and Katherine Lee for their support in data collection and processing.

This work was supported by grants NIH R01CA226527, NIH/NIGMS T32GM141746, NIH K12 NS080223, Cook Family Brain Tumor Research Fund, Mark Trauner Brain Research Fund: Zenkel Family Foundation, and Ian's Friends Foundation.

Research reported in this publication was also supported by the Investigators Awards grant program of Precision Health at the University of Michigan.

This research was also supported in part through computational resources and services provided by Advanced Research Computing (ARC), a division of Information and Technology Services (ITS) at the University of Michigan, Ann Arbor.

**Competing interests:** C.W.F. is an employee and shareholder of Invenio Imaging, Inc., a company developing SRH microscopy systems. D.A.O. is an advisor and shareholder of Invenio Imaging, Inc, and T.C.H. is a shareholder of Invenio Imaging, Inc.

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
