# OpenReview forum: "OpenSRH: optimizing brain tumor surgery using intraoperative stimulated Raman histology"
_NeurIPS.cc/2022/Track/Datasets_and_Benchmarks — NeurIPS 2022 Datasets and Benchmarks _

### Official Review · Reviewer_ZJAZ · 2022-07-21
**Worthwhile resource with detailed description and thorough experiments**

**Rating:** 9
**Confidence:** 4
**Correctness:** The dataset and the benchmark are wel…
**Clarity:** The paper is well-written, motivating…

**Strengths:**

- The published dataset opens up an interesting opportunity for optimizing brain tumor surgery through SRH imaging
- The established benchmark motivates future work to address the domain gap on the existing pretraining data and aggregation methods to better model slide-level and patient-level prediction
- The task described in the paper is well-motivated and well-written

**Weaknesses:**

**Patient-level prediction vs patient-level prediction** Despite the poor aggregation method is already briefly discussed on 6.2, I would expect there would be simple experiment to validate this phenomenon further by, for example, trying different aggregation methods during inference.

**Additional Feedback:**

- Appendix E: "except for the unfrozen model weights in finetuning experiments.", does the fine-tuning experiment refers to the section 5? I think this needs to be revised to make it clearer
- The domain gap cap also be described by explaining the gap from the linear evaluation and the full fine-tuning of the ImageNet models.

**Documentation:**

The paper provides sufficient details on the data collection and organization, availability, benchmark reproducibility, and all the ethical consequences.

**Ethics:**

The paper follows all ethical standards for responsible research practice and gets the approval from the Institutional Review Board.

**Relation To Prior Work:**

The paper clearly described prior works from other digital and computational pathology research and other SRH imaging studies.
The paper further explain what is required for future research and how the paper accommodates this requirement.

**Summary And Contributions:**

The paper introduces a new resource for brain optical imaging, namely OpenSRH, which is valuable for optimizing brain tumor surgery. The paper clearly explain the motivation of OpenSRH, which is easy enough to follow even without any prior knowledge in brain surgery or optical imaging. The paper provides sufficient details regarding the data collection methods and the dataset statistics. In addition, the paper develop two benchmarks from the dataset, i.e., histologic brain tumor classification and contrastive representation learning benchmarks. The experiment protocol for both benchmark is well-descibed and sufficient analysis is presented from the results.

---

> ### Author Response · Authors · 2022-08-26
> **Response to reviewer ZJAZ**
>
> We would like to thank the reviewer for the thoughtful and constructive feedback.
>
> **Weakness :** Patient-level prediction vs patient-level prediction Despite the poor aggregation method is already briefly discussed on 6.2, I would expect there would be simple experiment to validate this phenomenon further by, for example, trying different aggregation methods during inference.
>
> **Answer:** We agree with the reviewer that the aggregation method is an important area of research for SRH, and all patch based histology research. We opted for a simple aggregation method (soft voting) for these benchmarks, and achieved reasonable results. We will leave experiments to investigate aggregation for future work, and we have identified a few interesting directions. As described in section 7, future directions include expectation maximization, clustering, attention, and multiple instance learning.
>
> **Additional Feedback:** Appendix E: "except for the unfrozen model weights in finetuning experiments.", does the fine-tuning experiment refers to the section 5? I think this needs to be revised to make it clearer
>
> **Answer:**
> We also appreciate the reviewer’s feedback on the clarity of the appendix. The appendix is now Appendix F in the revision. We meant to convey that the finetuning protocol is the same as the linear evaluation protocol described in section 6.1, except for trainable (unfrozen) weights in the model backbone. We have updated the description to make our intent clearer.

---

### Official Review · Reviewer_6fG8 · 2022-07-24
**Reducing surgery wait-time via SRH and providing expert-level tumor classification important improvement**

**Rating:** 9
**Confidence:** 4

**Strengths:**

- manuscript very clearly lays out the advance made by switching from frozen sections to SRH imaging
- methods are well described and results well visualized; the main findings can be readily extracted: SRH reduces the time from sample extraction to usable information by about 30 minutes (75%); and using contrastive learning on a (frozen) computer vision model (head) significantly improves the classification accuracy of patch-level inference to (almost) patient-level accuracy (of the simple across-patch aggregated data); this second finding (patch-level inference matches simple-aggregate patient-level inference) might be highlighted in the abstract and discussion to make it stand out even more clearly
- data was collected for specific research/technology development interest, making it a very strong contribution to the field
- data and code is (or will be) shared for further collaboration and development


**Weaknesses:**

The only remaining question I had while reading occurred in lines 86-87 -- could the AI approach also be combined with light-microscopic (traditional) histopathology? it would seem important to separate the two contributions to the field (SRH as a speed up, and AI as a reduction in pathology expert reliance) for a more fine-grained evaluation of the claims; what would it take to train a similar AI on (high-resolution digitized) traditional H&E slides? And given how much a SRH scanner costs (and the lack of expert pathologist availability), is that the greater (or lesser) bottleneck in current practice?

**Additional Feedback:**

- this is a wonderful contribution to the literature, demonstrating state-of-the-art technology brought to the bedside; I congratulate the authors, and hope that more rapid and reliable intra-operative tumor diagnostic can soon become available for brain tumor patients everywhere
- figure 4: while maybe not entirely easy to achieve, it would be wonderful to see a confusion matrix on the patch level (specifically, how many patches of slides with non-normal tissue are mis-labeled), to get a sense of how necessary the aggregation into slide/patient-level classification is
- lines 248-249: would it be instructive (for future development) to show some of the slides (with classified as well as manually labeled patches from the cross-validation part of the training)? This might give readers some idea as to how a potentially improved aggregation function might look like
- line 258: for readers interested in applying this technique in their own setting, it might be beneficial to add (somewhere in the manuscript) the typical cost for obtaining the necessary equipment (SRH slide scanner, computer hardware necessary for inference, etc.), as well providing a clear path for collaboration (if already planned, maybe add a URL to the part of the opensrh.mlins.org site here?)
- figure 5: while definitely some extra work, it would be a great visualization to pick out some of the patches (across the different diagnoses among the validation images) and show where they "land" within the tSNE to help readers appreciate how the difference in "patterns" present in a patch might translate into the clustering location in the 2D projection


**Clarity:**

Only some **minor** points noted here:

- line 139, the unit (3x3 mm3) suggests either a planar or cubic specimen, please clarify
- lines 147 and 148: into how many such 300x300 patches would a typical slide be "cut" during preprocessing (to give readers a sense of the density of the underlying data, as well as over how much the aggregation function later needs to work)? have the authors tried different patch sizes (if so, what were the findings; if not, maybe it's worth considering, given that information as to tissue "class membership" may reside at different spatial frequencies/scales)? incidentally, what is the mm- (or micrometer) size of a 300x300 patch?
- plus, out of curiosity: have the authors tried (partially) overlapping windows? if not, why were non-overlapping windows chosen?
- lines 148-151: if the third channel is fully explained by channels 1 and 2, how does it add to the network performance? also, assuming that channels 1 and 2 are bound between 0 and some arbitrary upper value/unit (for full luminescence of the sensor, say, 1); the third channel would have double the width (between -1 and 1); how is that coded in the input of the data?
- line 153: who performed the manual labeling? (multiple experts? if so, what was their concordance? this might help further convince readers of the quality of the dataset)
- line 182: what function was used for the image resizing (linear/cubic interpolation, sinc/spline, etc.)?
- line 224: why does the number in dimensions for the hypersphere differ this much across the two model types (ResNet vs. ViT-S)?


**Correctness:**

From what I could tell, the claims seem correct, and I have no concerns at present; however, given that the data cannot currently be requested (the Google form states this will be at a later date), I was not able to inspect the dataset itself.

**Documentation:**

No issues with the description of the data or the site linked at the end of the abstract. However, as mentioned in "Correctness" section: currently the link to request data says (at the top of the Google form) that the data isn't available to (for) download yet; the reviewer was thus unable to perform spot checking of the data itself

**Ethics:**

No concerns.

**Relation To Prior Work:**

Since the authors already cite Dong Li's work (ref #3), maybe they would also like to include https://www.sciencedirect.com/science/article/pii/S1878875021003156 as a paper that already demonstrates the value of SRH specifically?

**Summary And Contributions:**

Overall, I have enjoyed this paper very much, and believe that it provides an excellent contribution to the literature/NeurIPS.

Major points:
- rapid intra-operative histopathological assessment of tumor samples seems incredibly desirable/valuable
- two aspects are highlighted: increased speed (savings of approx. 30 minutes/75% of time after a sample is taken and transferred to the slide) and through digitization and AI/computer vision techniques a reduced dependence on human neuropathology expert opinion; it would be important to test this with a pre-clinical prospective study (what is the influence of AI generated labels/diagnoses on surgical decision making; what are benefits/harms compared to current standard of care; when and how can over-reliance on AI lead to detrimental outcomes, etc.)
- the dataset, together with the workflow/code from the authors will allow rapid development of these protocols/studies/trials to demonstrate the clinical value of SRH as an integral part of tumor surgery, and given the open access (upon request, as it seems) will allow other groups to increase the speed of adoption of this technology

---

> ### Author Response · Authors · 2022-08-26
> **Response to reviewer 6fG8 (part 1)**
>
> We would like to thank the reviewers for the thoughtful and constructive feedback. To address the reviewer’s comments:
>
> **Weakness 1:** The only remaining question I had while reading occurred in lines 86-87 -- could the AI approach also be combined with light-microscopic (traditional) histopathology? it would seem important to separate the two contributions to the field (SRH as a speed up, and AI as a reduction in pathology expert reliance) for a more fine-grained evaluation of the claims; what would it take to train a similar AI on (high-resolution digitized) traditional H&E slides?
>
> **Answer:** Our AI approach could be applied to light microscopy and traditional histopathology. The methods described in this paper such as patch-based contrastive learning can be applied to any histopathology dataset such as TCGA with little to no modification.
>
> **Weakness 2:** And given how much a SRH scanner costs (and the lack of expert pathologist availability), is that the greater (or lesser) bottleneck in current practice?
>
> **Answer:** Our hope in releasing OpenSRH is to facilitate AI research in both SRH and conventional H&E. Currently, the cost associated with an SRH scanner is less than maintaining a histopathology laboratory, and when combined with AI, reduces the reliance on expert pathologists for intraoperative diagnosis.

---

> ### Author Response · Authors · 2022-08-26
> **Response to reviewer 6fG8 (part 2)**
>
> **Clarity 1:** line 139, the unit (3x3 mm3) suggests either a planar or cubic specimen, please clarify
>
> **Answer:** The specimen is volumetric (3D) and placed in the center of the microscope slide, then it is covered by another glass flap built into the slide. Then the strips are acquired using optical sectioning at a single depth to generate a planar image.
>
> **Clarity 2a:** lines 147 and 148: into how many such 300x300 patches would a typical slide be "cut" during preprocessing (to give readers a sense of the density of the underlying data, as well as over how much the aggregation function later needs to work)?
>
> **Answer:** The number of patches varies from slide to slide since the size for a slide can be varied as well. Our most frequent slide sizes are 6000x6000. And usually 324 patches can be generated from such size.
>
> **Clarity 2b:** have the authors tried different patch sizes (if so, what were the findings; if not, maybe it's worth considering, given that information as to tissue "class membership" may reside at different spatial frequencies/scales)?
>
> **Answer:** Smaller patch sizes (256) has been tried for different experiments especially in generative learning tasks, and these patch sizes are chosen because of 1) storage and model training convenience, as some architectures require a fixed input size; and 2) patch size of 300 evenly divides the strip width with a 50 pixel margin on each side. The edges of the strips overlap each other and can have higher laser noise due to lower laser power and are discarded. In our experience, cellular features are always smaller than patch sizes, but some extracellular structures such as blood vessels may span multiple patches.
>
> **Clarity 2c:** incidentally, what is the mm- (or micrometer) size of a 300x300 patch?
>
> **Answer:** A 300x300 pixel patch captures a field of view of size 133 microns squared.
>
> **Clarity 3:** plus, out of curiosity: have the authors tried (partially) overlapping windows? if not, why were non-overlapping windows chosen?
>
> **Answer:** Previous study [1] of our group tried sampling data with overlapping windows. The overlapping window has its usage for visualization as segmentation heatmap. Since this dataset is mainly designed for histological classification, we believe overlapping strategies can increase data redundancy and computational cost and non-overlapping windows is a more efficient way to train the model.
>
> **Clarity 4:** lines 148-151: if the third channel is fully explained by channels 1 and 2, how does it add to the network performance? also, assuming that channels 1 and 2 are bound between 0 and some arbitrary upper value/unit (for full luminescence of the sensor, say, 1); the third channel would have double the width (between -1 and 1); how is that coded in the input of the data?
>
> **Answer:** We added the third channel image because three-channel images are 1) easy to visualize with RGB channels; and 2) friendly to transfer learning (e.g., using ImageNet pretrained models and weights directly).  It is true that neural networks can easily learn this subtraction relationship (or a 3 channel representation), but that would require additional learned parameters, larger memory footprint and a longer training time. Each channel is rescaled to between 0 and 1 before processing, and we add a base to the subtracted channel so that these pixel values are positive. Implementation details are included in the companion source code on [Github](https://github.com/MLNeurosurg/opensrh/blob/265591195d700ab8950d17b45f50dd2467fa02a4/opensrh/datasets/improc.py#L94).
>
> **Clarity 5:** line 153: who performed the manual labeling? (multiple experts? If so, what was their concordance? this might help further convince readers of the quality of the dataset)
>
> **Answer:** The ground truth of the whole-slide and patient classification task as well as segmented patch labels were performed by board-certified neuropathologists.
>
> **Clarity 6:** line 182: what function was used for the image resizing (linear/cubic interpolation, sinc/spline, etc.)?
>
> **Answer:** The image resizing transformation is performed using the torchvision python package with the default bilinear interpolation option.
>
> **Clarity 7:** line 224: why does the number in dimensions for the hypersphere differ this much across the two model types (ResNet vs. ViT-S)?
>
> **Answer:** The number of dimensions of different architectures is different (2048 in ResNet50 and 384 in ViT-S), and we chose the output dimension of the hypersphere / projection layer to be 1/16 of the model output embeddings.
>
> **Clarity 8:** Relation To Prior Work: Since the authors already cite Dong Li's work (ref #3), maybe they would also like to include https://www.sciencedirect.com/science/article/pii/S1878875021003156 as a paper that already demonstrates the value of SRH specifically?
>
> **Answer:** We apologize for our inadvertent omission of this article. We have added this as a reference to the manuscript.

---

> ### Author Response · Authors · 2022-08-26
> **Response to reviewer 6fG8 (part 3)**
>
> **Additional Feedback 1:** figure 4: while maybe not entirely easy to achieve, it would be wonderful to see a confusion matrix on the patch level (specifically, how many patches of slides with non-normal tissue are mis-labeled), to get a sense of how necessary the aggregation into slide/patient-level classification is
>
> **Answer:** We have included patch, slide, and patient level confusion matrix for seed 1 of histological classification in Figure 13 in appendix D. These confusion matrices correspond to the same experiments in Figure 1.
>
> **Additional Feedback 2:** lines 248-249: would it be instructive (for future development) to show some of the slides (with classified as well as manually labeled patches from the cross-validation part of the training)? This might give readers some idea as to how a potentially improved aggregation function might look like
>
> **Answer:** Our team has been implementing a demonstration [website](deepglioma.mlins.org) for interactive inspection of SRH whole images with associated annotations and prediction. Interactive website is currently focused on gliomas, but extension to other tumor types is planned.
>
> **Additional Feedback 3:** line 258: for readers interested in applying this technique in their own setting, it might be beneficial to add (somewhere in the manuscript) the typical cost for obtaining the necessary equipment (SRH slide scanner, computer hardware necessary for inference, etc.), as well providing a clear path for collaboration (if already planned, maybe add a URL to the part of the opensrh.mlins.org site here?)
>
> **Answer:** The cost to acquire a SRH scanner is approximately $400K USD. We have included a “Contact” page on opensrh.mlins.org for anyone who would like to collaborate with us.
>
> **Additional Feedback 4:** figure 5: while definitely some extra work, it would be a great visualization to pick out some of the patches (across the different diagnoses among the validation images) and show where they "land" within the tSNE to help readers appreciate how the difference in "patterns" present in a patch might translate into the clustering location in the 2D projection
>
> **Answer:** Thank you for the useful suggestion! In the interest of manuscript length, we are developing a fully interactive tSNE data visualization tool that includes hover function and image visualization for the website. [**Here**](https://youtu.be/SVBr7GcXY3A) is a sneak peak of the tool running locally.
>
> **References:**
>
> [1] Todd C Hollon, Balaji Pandian, Arjun R Adapa, Esteban Urias, Akshay V Save, Siri Sahib S Khalsa, Daniel G Eichberg, Randy S D’Amico, Zia U Farooq, Spencer Lewis, Petros D Petridis, Tamara Marie, Ashish H Shah, Hugh J L Garton, Cormac O Maher, Jason A Heth, Erin L McKean, Stephen E Sullivan, Shawn L Hervey-Jumper, Parag G Patil, B Gregory Thompson, Oren Sagher, Guy M McKhann, 2nd, Ricardo J Komotar, Michael E Ivan, Matija Snuderl, Marc L Otten, Timothy D Johnson, Michael B Sisti, Jeffrey N Bruce, Karin M Muraszko, Jay Trautman, Christian W Freudiger, Peter Canoll, Honglak Lee, Sandra Camelo-Piragua, and Daniel A Orringer. Near real-time intraoperative brain tumor diagnosis using stimulated raman histology and deep neural networks. *Nat. Med.,* January 2020.

---

### Official Review · Reviewer_8qJ4 · 2022-07-25
**A 300+ patients brain tumor dataset with benchmark**

**Rating:** 6
**Confidence:** 4
**Correctness:** The claims made in the paper seem to …

**Strengths:**

- The authors well documented the need for this dataset through their prior work and sound arguments regarding the specifications of an ideal system for brain tumor surgeries.

- The paper included two well-documented and explained benchmarks that exemplify their proposed workflow and the significance of the dataset. Additionally, enough details are provided for reproducibility.

**Weaknesses:**

- The paper recommends a workflow for classifying brain tumors using Machine Learning instead of existing resource-intensive workflows. However, the paper doesn't touch on the accuracy of their automated process compared to the existing anywhere in the paper. I think a brief mention of this is needed as a current limitation if there is a dip in accuracy or a strong point if the accuracy is improved.

- The paper didn't highlight or explicitly elaborate on their results in both tables (1 - 2), although I think the results are rich enough for a rich discussion and comparison of the results.

**Additional Feedback:**

N/A

**Clarity:**

- Overall, the reading flow of this paper was easy. Highlighted bullet points with meaningful descriptions of what to expect in the following paragraphs were beneficial. An excellent example of this is section (4) data description.

In section 5.3, the authors mentioned, "We did observe improved performance when using ImageNet pretraining." However, based on table 1, the patient-level accuracy does not follow this observation. A more detailed analysis is needed.

In Tables 1 and 2, the authors may make the best result in each column bold.

**Documentation:**

Dataset and code are publicly available, and the links are provided in the paper. The researchers also provide a website that integrates the paper, code, and dataset.

**Ethics:**

This paper follows the ethics guidelines.

**Relation To Prior Work:**

The paper provides detailed background knowledge about SRH and its applications in brain tumor surgeries, allowing non-experts to understand this paper's significance quickly.
Previous related works are clearly explained and smoothly embedded into this paper. The authors discussed the current difficulties in ML for SRH.

**Summary And Contributions:**

The paper introduces OpenSRH,  a public dataset including 1300+ clinical SRH images from 300+ brain tumor patients, and a benchmark for two computer vision tasks: multiclass histologic brain tumor classification and patch-based contrastive representation learning.
The proposed workflow demonstrated in the benchmarking, and the dataset aims to accelerate the development of fast, reliable, and accessible intraoperative diagnosis practices for brain tumor surgeries.

---

> ### Author Response · Authors · 2022-08-26
> **Response to reviewer 8qJ4**
>
> We are thankful for the reviewer’s thoughtful and constructive feedback! To address the reviewer’s comments:
>
> **Weakness 1:** The paper recommends a workflow for classifying brain tumors using Machine Learning instead of existing resource-intensive workflows. However, the paper doesn't touch on the accuracy of their automated process compared to the existing anywhere in the paper. I think a brief mention of this is needed as a current limitation if there is a dip in accuracy or a strong point if the accuracy is improved.
>
> **Answer:** We apologize for not including existing workflow metrics in the paper. Conventional pathology workflow with board certified neuropathologist interpretation has a classification accuracy between 86 - 96% [1], and there are no automated or decision support tools using ML methods currently in clinical practice for brain tumor diagnosis. We have included this information in section 1 in the revised manuscript.
>
> **Weakness 2:** The paper didn't highlight or explicitly elaborate on their results in both tables (1 - 2), although I think the results are rich enough for a rich discussion and comparison of the results.
>
> **Answer:** The reviewer made a good point in the clarity of the results. We have elaborated on the results reported in tables 1 and 2 in sections 5.3 and 6.2.
>
> **Clarity 1:** In section 5.3, the authors mentioned, "We did observe improved performance when using ImageNet pretraining." However, based on table 1, the patient-level accuracy does not follow this observation. A more detailed analysis is needed.
>
> **Answer:** We apologize for being ambiguous when discussing the metrics. In the statement you quoted, we were referring to patch level performance, which we observed a significant boost in accuracy with ImageNet pre-training. As you correctly pointed out, the patient accuracy for ResNet50 did not improve, but is within the margin of error. We have updated the text in section 5.3 to reflect the improved performance to be in patch-level metric only.
>
> **Clarity 2:** In Tables 1 and 2, the authors may make the best result in each column bold.
>
> **Answer:** We would like to thank the reviewer for the suggestion! We have highlighted numerical results in tables 1 and 2 in the revised manuscript.
>
> **References :**
>
> [1] Daniel A Orringer, Balaji Pandian, Yashar S Niknafs, Todd C Hollon, Julianne Boyle, Spencer Lewis, Mia Garrard, Shawn L Hervey-Jumper, Hugh J L Garton, Cormac O Maher, Jason A Heth, Oren Sagher, D Andrew Wilkinson, Matija Snuderl, Sriram Venneti, Shakti H Ramkissoon, Kathryn A McFadden, Amanda Fisher-Hubbard, Andrew P Lieberman, Timothy D Johnson, X Sunney Xie, Jay K Trautman, Christian W Freudiger, and Sandra Camelo-Piragua. Rapid intraoperative histology of unprocessed surgical specimens via fibre-laser-based stimulated raman scattering microscopy. *Nat Biomed Eng,* 1, February 2017.

---

### Official Review · Reviewer_cLcj · 2022-07-26
**Very interesting on the medical front. Not sure about the machine learning side.**

**Rating:** 5
**Confidence:** 3

**Strengths:**

The medical problem is well stated and seems important.
The github repository is well conceived and user-friendly while the dataset is easy to access and well constructed.
The approach of the tumour segmentation and classification tasks is rigorous.


**Weaknesses:**

As mentioned above, the reviewer does question the added value of the dataset to ML's innovation.


**Additional Feedback:**

NA

**Clarity:**

The paper is very clear and well written. The medical objective is well explained for non-specialists in medical imaging.
The supplementary material is well conceived and very useful.
Access to the dataset and the github repo are user-friendly.


**Correctness:**

The dataset seems constructed is a very sound as well as innovative way.
Benchmarks for the studied segmentation and classification tasks and contrastive learning techniques are classical and seem pretty natural.


**Documentation:**

The main paper, the supplementary and the github repo do provide enough information on the dataset building process and the licensing.
A date use agreement is being finalised by the authors.


**Ethics:**

Contribution to a positive societal impact is obvious.
No potential privacy breach for patients in the dataset was detected by the reviewer.


**Relation To Prior Work:**

The differentiation of this dataset vs. other histopathology repositories is pretty appealing and well explained.

**Summary And Contributions:**

The medical problem being tackled in this paper seems exciting and novel (given the reviewer's knowledge about medicine which is not much). The SRH dataset is apparently unique and would allow real-time brain tumour segmentation and classification with significant consequences for patients.
This being said, the reviewer is not convinced that this paper suits the NeurIPS conference's or the NeurIPS Datasets & Benchmarks Track's objectives.
While the medical added value of the paper could be interesting (from what the reviewer can only guess), the interest for the machine learning community seems pretty limited: the dataset is a medical imaging repository and the techniques being benchmarked are very classical computer vision models and contrastive learning methods. It is not clear to the reviewer that this new dataset will foster particularly innovative discoveries on the machine learning front.

---

> ### Author Response · Authors · 2022-08-26
> **Response to reviewer cLcj**
>
> We would like to thank the reviewer for the thoughtful and constructive feedback! To address the reviewer’s comments:
>
> **Weakness:** This being said, the reviewer is not convinced that this paper suits the NeurIPS conference's or the NeurIPS Datasets & Benchmarks Track's objectives. While the medical added value of the paper could be interesting (from what the reviewer can only guess), the interest for the machine learning community seems pretty limited: the dataset is a medical imaging repository and the techniques being benchmarked are very classical computer vision models and contrastive learning methods. It is not clear to the reviewer that this new dataset will foster particularly innovative discoveries on the machine learning front.
>
> **Answer:** We apologize for not making clear the relevance of the dataset to the ML community. The open questions for the ML community that OpenSRH may foster innovative discoveries include the following: 1) domain adaptation between SRH images and other histology images such as H&E images in the large scale TCGA project; 2) using multiple instance learning (MIL) to avoid expensive dense patch annotations; 3) different aggregation methods for patch-based training, including clustering, attention, or MIL; 4) self supervised learning and comparing different augmentation strategies for SRH images; and 5) data efficient training of ViT architectures using SRH data. We have modified section 7 to further address the relevance to the ML community and open questions.

---

### Official Review · Reviewer_xEv2 · 2022-07-27
**First public dataset of stimulated Raman histology images of brain tumors**

**Rating:** 8
**Confidence:** 4
**Correctness:** Yes, the claims made in the submissio…

**Strengths:**

The authors provide strong motivation for their work -- specifically that the pathologic diagnosis of brain tumors is usually unknown until surgery, whereas it is known prior to surgery for other cancers. The authors also successfully motivate the use of stimulated Raman histology over hematoxylin and eosin-stained histology (such as frozen sections that might be used for diagnosis during surgery). Another strength is in the description of stimulated Raman histology and SRH imaging. Moreover, the data and code are all available (to reviewers at the moment, and to the public in the near future), and the appendix provides meticulous documentation of the data and training.

**Weaknesses:**

There are very few weaknesses I can identify in this work. The authors have done an excellent job. One weakness I would like to point out is in the evaluation metrics. The authors use accuracy (and variations) and mean average precision. In addition to these, I highly suggest including a metric that highlights false negatives. False negatives would be unacceptable in a diagnostic task, so the authors should provide benchmark evaluation of this and highlight the amount of false negatives.

In addition, I would caution against using accuracy as an evaluation metric in this case, because the labels are unbalanced (based on the number of slides in Figure 3). The authors should report metrics that properly handle unbalanced data, like AUROC and AUPR (or related metrics).

**Additional Feedback:**

The authors indicate that data will be available via Google Drive and Amazon AWS S3. In my experience, downloading data from Google Drive has been inconvenient, because it does not allow for programmatic downloads (e.g., with wget or curl on a remote machine). I would encourage the authors to consider a different hosting mechanism. Perhaps DropBox, Zenodo, or an AWS S3 bucket that does not require an account.

Can the authors indicate the range of patient ages in the dataset? Some brain tumors are much more common in children, while others (like metastases) would be more likely in adults. It is possible that the tissue or tumor microenvironment of a younger patient could be different than that of an older patient, and this is why I am asking about the range of ages.

In lines 125-126, the authors write that "SRH image acquisition is invariant to patient demographic features, clinical workforce, and geographic location." Can the authors please explain this claim? How are they certain that SRH image acquisition is invariant to all of these factors?

Regarding partitioning data into train/val, are the data split by patient or by slide? The data should be split by patient, so that no patient is present in more than one data split (otherwise there could be leakage between splits). --- After reading Figure 9 in the Appendix, I see that the data is split by patient. But I will let the authors confirm that this is indeed the case.

On line 145, the authors indicate that the two channels of an SRH image are co-registered. How robust is this co-registration? Does it every fail?

On lines 200-201, the authors write that ViT requires larger data than ImageNet-1k --- I agree with this. In the future, the authors can consider evaluating transfer learning from a vision transformer pretrained on ImageNet-22K (they are publicly available).

**Clarity:**

Yes, the paper is very well written. There is one section that I did not understand fully (lines 247-249): "Interestingly, the patient-level metrics for pretrained ViT-S were prominently high, while the patch-level metrics were comparatively worse. We believe these results may be due to poor aggregation of patch-level predictions." If there is poor aggregation of patch-level predictions, wouldn't one expect poor patient-level predictions as well? Perhaps I am misunderstanding here. Can the authors please clarify?

**Documentation:**

Overall, yes. The data is documented well. In terms of availability, the authors have indicated that they are working on a data use agreement. Can the authors indicate whether the data will be publicly available in the near future (approximate date)? I also mentioned in "Additional Feedback" that the authors should consider a hosting platform other than Google Drive (because it does not allow for programmatic downloads, like with curl/wget). If the data were hosted on a platform like Zenodo, I would not have any concerns about hosting or maintenance. If the data is to be hosted on in a privately managed space like a Google Drive directory or AWS S3 bucket, can the authors please explain their plans for maintaining the data online and maintaining access to it?

**Ethics:**

This dataset contains clinical images from patients, and the authors have removed private health information from the data.

**Relation To Prior Work:**

The authors provide a great summary of previous work. Indeed, much of the related prior work is in hematoxylin and eosin-stained histology images. One large claim that the authors make is that OpenSRH is the first public SRH dataset for brain tumors. I cannot verify this because I have insufficient knowledge of existing SRH datasets.

**Summary And Contributions:**

The authors introduce a public dataset of stimulated Raman histology images of brain tumors (OpenSRH). The authors claim that this is the first dataset of its kind (I cannot confirm or deny this, because I am not familiar with previous literature in Raman histology datasets). This dataset includes over 1,300 whole slide images from 307 patients. The authors also establish classification and representation learning benchmarks on this dataset. This is a rich dataset, which includes raw acquired data, patch-wise tumor classifications for the whole slide images, and processed patches ready for ML pipelines. Overall, this is an excellent and exciting contribution to the field.

---

> ### Author Response · Authors · 2022-08-26
> **Response to reviewer xEv2 (part 1)**
>
> We thank the reviewer for the support and careful reading of our manuscript. We found the feedback constructive. To address the reviewer’s comments:
>
> **Weakness 1:** The authors use accuracy (and variations) and mean average precision. In addition to these, I highly suggest including a metric that highlights false negatives. False negatives would be unacceptable in a diagnostic task, so the authors should provide benchmark evaluation of this and highlight the amount of false negatives.
>
> **Answer:** The reviewer makes an excellent point regarding the importance of false negative diagnostic errors. We have included a statement in the manuscript regarding false negative diagnoses and their importance. We have added the false negative rates as an additional metric to each metrics table in the appendices.
>
> **Weakness 2:** In addition, I would caution against using accuracy as an evaluation metric in this case, because the labels are unbalanced (based on the number of slides in Figure 3). The authors should report metrics that properly handle unbalanced data, like AUROC and AUPR (or related metrics).
>
> **Answer:** The reviewer correctly states that raw accuracy does not account for class imbalance. To address class imbalance, we have included mean class accuracy, which averages accuracy for each class and is sensitive to class imbalance. We have also updated the result tables in the manuscript to include AUPR (mean average precision).
>
> **Clarity:**  There is one section that I did not understand fully (lines 247-249): "Interestingly, the patient-level metrics for pretrained ViT-S were prominently high, while the patch-level metrics were comparatively worse. We believe these results may be due to poor aggregation of patch-level predictions." If there is poor aggregation of patch-level predictions, wouldn't one expect poor patient-level predictions as well? Perhaps I am misunderstanding here. Can the authors please clarify?
>
> **Answer:**  We apologize for the lack of clarity with this statement on lines 247-249. We meant to convey that a simple aggregation (soft voting) method is used, and this has the potential to result in suboptimal patient level predictions. The simple method can achieve good results over all, but better aggregation methods should be explored in future work. We have edited the manuscript to improve clarity of this sentence.

---

> ### Author Response · Authors · 2022-08-26
> **Response to reviewer xEv2 (part 2)**
>
> **Additional Feedback 1:** The authors indicate that data will be available via Google Drive and Amazon AWS S3. In my experience, downloading data from Google Drive has been inconvenient, because it does not allow for programmatic downloads (e.g., with wget or curl on a remote machine). I would encourage the authors to consider a different hosting mechanism. Perhaps DropBox, Zenodo, or an AWS S3 bucket that does not require an account.
>
> **Answer:** The reviewer has made a good point that downloading from google drive might not be best. Hence, we are also making this data available via Dropbox; the link will be available promptly after the data usage agreement is signed.
>
>
> **Additional Feedback 2:** Can the authors indicate the range of patient ages in the dataset? Some brain tumors are much more common in children, while others (like metastases) would be more likely in adults. It is possible that the tissue or tumor microenvironment of a younger patient could be different than that of an older patient, and this is why I am asking about the range of ages.
>
> **Answer:** In regards to the range of patient ages, we have included a histogram of patient ages by tumor diagnosis in the Appendix B. We observed that high grade gliomas, metastasis, and meningiomas are more common in older patients, while low grade gliomas are more common in younger patients.
>
> **Additional Feedback 3:** In lines 125-126, the authors write that "SRH image acquisition is invariant to patient demographic features, clinical workforce, and geographic location." Can the authors please explain this claim? How are they certain that SRH image acquisition is invariant to all of these factors?
>
> **Answer:** With regards to SRH image acquisition, the current SRH imagers (NIO Imaging System, Invenio Imaging, Inc.) distributed across the US and Europe have the same fiber-laser and image acquisition specifications with quality assurance. Additionally, no tissue processing, such as staining or dyeing, is required for image acquisition.
>
> **Additional Feedback 4:** Regarding partitioning data into train/val, are the data split by patient or by slide? The data should be split by patient, so that no patient is present in more than one data split (otherwise there could be leakage between splits). --- After reading Figure 9 in the Appendix, I see that the data is split by patient. But I will let the authors confirm that this is indeed the case.
>
> **Answer:** Yes, the data is randomly split by patients.
>
> **Additional Feedback 5:** On line 145, the authors indicate that the two channels of an SRH image are co-registered. How robust is this co-registration? Does it every fail?
>
> **Answer:** We used [Fast Fourier Transform](https://imreg-dft.readthedocs.io/en/latest/) to coregister our two channel images. This has been a robust method for medical image coregistration tasks. We have not observed significant failures of the registration in tissue, however we have seen minor errors in empty, nondiagnostic spaces.
>
> **Additional Feedback 6:** On lines 200-201, the authors write that ViT requires larger data than ImageNet-1k --- I agree with this. In the future, the authors can consider evaluating transfer learning from a vision transformer pretrained on ImageNet-22K (they are publicly available).
>
> **Answer:** We would like to thank the reviewer for the suggestion! Pretraining with ImageNet-22K is a worthwhile experiment that we will perform in the future! Any results / updates from these experiments will be posted on our [website](https://opensrh.mlins.org).

---

> > ### Comment · Reviewer_xEv2 · 2022-08-29
> > **Thank you**
> >
> > Thank you for your comprehensive responses. I have no further questions. Great work!

---

### Official Review · Reviewer_YwAc · 2022-07-27
**NeurIPS 2022 Track Datasets and Benchmarks Paper264 Reviewer YwAc**

**Rating:** 3
**Confidence:** 4
**Clarity:** Yes.

**Strengths:**

* The community needs more medical imaging data sets, and this is one covers a large number of tumor types and patients.
* The benchmarking seems of very high quality and is very thoroughly described.

**Weaknesses:**

* The "ground truth" dataset itself was categorized based on a neural network. This is not a problem in general, but the description of exactly how this was done is by far the weakest part of the paper. Who were they manually checked by? How much of the data was checked at each stage? It is hard to evaluate any of the results if the ground truth is not certain, and in a paper proposing a data set and that it should be used as a benchmark it is critical to understand this.
* The data comes from a single institution and a single commercial instrument. It is not clear to this reviewer whether one or many companies produce such instruments, but if this data only helps purchasers of a single (or a few) commercial instruments, the breadth of its utility seems limited to me.
* There is not a current finalized data use agreement. Exactly how "open" OpenSRH will be cannot be determined until this is true.
* The data is currently structured as either a single extremely large tarball (>300GB) or per-patient. The latter is appreciated, but without information about which patient(s) have which tumor types, it isn't clear how would would know which patients to sub-sample. The metadata.json files are not available as far as I can tell except in the large tarball.

**Additional Feedback:**

None

**Correctness:**

The benchmark evaluation and experimental design appear correct to me. Per the "Weaknesses" section, the dataset does not contain enough information to assess this.

**Documentation:**

* Hosting and licensing are described, but not a maintenance plan. Ethical and responsible use are not sufficiently described.
* The benchmarking of the data appears sufficiently described.

**Ethics:**

No information is provided about the patient samples, so it is impossible to know if such aspects as race or gender are accidentally encoded. Other ethical concerns appear to be addressed.

**Relation To Prior Work:**

Yes.

**Summary And Contributions:**

This paper provides a data set of SRH images from a number of types of brain tumors as well as control tissue. It also provides several benchmarks for accuracy of classification of these images at the individual patch, slide, and patient level.

---

> ### Author Response · Authors · 2022-08-26
> **Response to reviewer YwAc**
>
> We would like to thank the reviewer for the thoughtful and constructive feedback. To address each weakness the reviewer commented:
>
> **Weakness 1:** The "ground truth" dataset itself was categorized based on a neural network. This is not a problem in general, but the description of exactly how this was done is by far the weakest part of the paper. Who were they manually checked by? How much of the data was checked at each stage? It is hard to evaluate any of the results if the ground truth is not certain, and in a paper proposing a data set and that it should be used as a benchmark it is critical to understand this.
>
> **Answer:** We apologize for the confusion regarding the ground truth whole-slide labels and the benchmarked classification task. The task we are benchmarking is whole-slide and patient brain tumor classification. The ground truth of the whole-slide and patient classification task is determined by the clinical tumor diagnosis performed by a board-certified neuropathologist. We used a segmentation model to identify tumor infiltrated regions in the whole slide images. Our aim in this paper was not to benchmark tumor segmentation; however, the whole slide segmentations are also released as part of the data for transparency. The detailed description of the segmentation model has been described in Appendix A.
>
> **Weakness 2:** The data comes from a single institution and a single commercial instrument. It is not clear to this reviewer whether one or many companies produce such instruments, but if this data only helps purchasers of a single (or a few) commercial instruments, the breadth of its utility seems limited to me.
>
> **Answer:** This is an important point made by the reviewer. Although we are the first institution to release public SRH data, SRH is being actively used for clinical diagnosis at other institutions in the United States and Europe. The data release in OpenSRH was generated from two SRH imagers that are produced by a single company (Invenio Imaging, Inc., Santa Clara, CA). Other related imaging technologies, such as optical coherence tomography, confocal microscopy, and second-harmonic generation imaging, can benefit from OpenSRH.
>
> **Weakness 3:** There is not a current finalized data use agreement. Exactly how "open" OpenSRH will be cannot be determined until this is true.
>
> **Answer:** This is a valid concern made by the reviewer. We want to make sure that there are no possibilities of using SRH images for commercial development without the permission of University of Michigan and Invenio Imaging. The data use agreement is now finalized and available on the  [website](https://opensrh.mlins.org).
>
> **Weakness 4:** The data is currently structured as either a single extremely large tarball (>300GB) or per-patient. The latter is appreciated, but without information about which patient(s) have which tumor types, it isn't clear how would would know which patients to sub-sample. The metadata.json files are not available as far as I can tell except in the large tarball.
>
> **Answer:** The data is available via Dropbox and Google Drive with each patient zipped into their own archive file. The data is also available on AWS without archiving. The metadata file `opensrh.json` is available separately.

---

### Meta-Review · Area_Chair_4XFE · 2022-09-10

**Recommendation:** Accept
**Confidence:** 4

**Metareview:**

There are contrasting review rating ranging from 9 to 3. The reasonable concerns about this paper not been well validated for ML community. The proposed datasets has been benchmarked with some classical and some relevant DL methods. I have taken into account the expertise of reviewers and their concerns carefully. I do agree with some reviewers opinion about this dataset will be importance to medical CV community. I do agree with authors response "The open questions for the ML community that OpenSRH may foster innovative discoveries include the following: 1) domain adaptation between SRH images and other histology images such as H&E images in the large scale TCGA project; 2) using multiple instance learning (MIL) to avoid expensive dense patch annotations; 3) different aggregation methods for patch-based training, including clustering, attention, or MIL; 4) self supervised learning and comparing different augmentation strategies for SRH images; and 5) data efficient training of ViT architectures using SRH data. " Even though the authors have not verified these innovative applications on the openSRH dataset. I will go with acceptance opinion of few reviewers for this paper.

---

### Decision · Program_Chairs · 2022-09-16

Accept